# Breeding displacement in gray wolves *(Canis lupus)*: Three males usurp breeding position and pup rearing from a neighboring pack in Yellowstone National Park

**Jeremy SunderRaj**[1]*, **Jack W. Rabe**[1,2], **Kira A. Cassidy**[1], **Rick McIntyre**[1], **Daniel R. Stahler**[1], **Douglas W. Smith**[1]

**1** Yellowstone Wolf Project, Yellowstone Center for Resources, Yellowstone National Park, Wyoming, United States of America, **2** Department of Fisheries, Wildlife, and Conservation Biology, University of Minnesota, St. Paul, Minnesota, United States of America

* j.sunderraj@hotmail.com

## Abstract

Gray Wolves *(Canis lupus)* are territorial, group living carnivores that live in packs typically consisting of a dominant breeding pair and their offspring. Breeding tenures are relatively short and competitive, with vacancies usually occurring following a breeder's death, and are often filled by unrelated immigrants or by relatives of the previous breeder. The frequency and conditions of active breeder displacements are poorly understood. Position changes in the dominance hierarchy are common yet rarely documented in detail. We describe a male breeding position turnover in a wolf pack by males from a neighboring pack in mid-summer 2016 in Yellowstone National Park. Over the course of two months, three males from the Mollie's pack displaced the breeding male of the neighboring Wapiti Lake pack, joined the pack's two adult females, and subsequently raised the previous male's four approximately three-month old pups. In the five years following the displacement (2017 to 2021), at least one of the intruding males has successfully bred with the dominant female and most years with a subordinate female (who was one of the pups at the time of displacement). The pack reared pups to adulthood each year. Male breeding displacements are likely influenced by male-male competition and female mate choice. These changes are the result of individuals competing to improve breeding position and may lead to increased pack stability and greater reproductive success. We report in detail on the behavior of a closely observed breeding displacement and we discuss the adaptive benefits of the change.

## Introduction

The opportunity to breed is an important driver in the behavior of all species, including large mammals [1]. However, many species of mammals are relatively short-lived, resulting in a more constrained timeframe to breed [2]. Access to breeding opportunities, therefore, are often met with conflict and reproductive competition [3]. As a result, breeding positions in

**Funding:** Funding was provided by the National Park Service (https://www.nps.gov/index.htm), Yellowstone Forever (https://www.yellowstone.org/), and the National Science Foundation (DEB-06137730, DEB-1245373; https://www.nsf.gov/). The funders had no role in study design, data collection and analysis, decision to publish, or preparation of the manuscript.

**Competing interests:** The authors have declared that no competing interests exist.

social mammals often have high turnover rates [4]. Reproductive competition is observed in a broad range of taxa (e.g., social carnivores, primates, ungulates, and pinnipeds), with reproductive skew resulting from individuals vying for dominance to maximize reproductive success [5]. Changes in mammalian breeding status can occur with one male usurping another, as intrasexual competition tends to be more severe among males than females; however, females are often the limiting sex and female mate choice may ultimately decide the success of male-male competition and is driven by physical, genetic, behavioral, and situational cues [6]. Mate preference is thought to benefit individuals most likely to bring the greatest direct and indirect fitness to the offspring [5]. Breeding competition and, perhaps paradoxically, sociality can be highly advantageous and essential to the survival of the group members, with benefits ranging from hunting to territorial disputes to cooperative breeding, leading to genetic diversity and pack stability [7–10].

Wolves *(Canis lupus)* are social animals that can exhibit complex and dynamic dominance hierarchies, ranging from simple linear structures to sex/age graded structures [11]. In some areas wolves demonstrate sex-specific breeding strategies, with females more likely to obtain breeding positions through natal philopatry (breeding in the pack they were born into, either through positional inheritance from a same-sex relative or by becoming a subordinate breeder), while males typically become breeders through dispersal (filling a breeder vacancy, usurping a dominant breeder) and temporary non-pack female affiliations [6]. These strategies, with males more likely to disperse and females to remain in their natal pack, can lead to matrilineal pack structures [12]. As a result, most wolf packs consist of a male and female adult who are unrelated to each other, their offspring from one or more years, and sometimes other adults that are related to either the dominant male or female [13]. The highest-ranking individuals in wolf packs usually breed. Obtaining a breeding position, especially in packs where multiple breeders occur, can lead to aggressive competition [10, 14, 15]. In captivity, old dominant males have been killed by their male offspring [14, 16], but such aggression is relatively rare in the wild [15, 17].

Most studies that report intense competition for breeding dominance in wolves are from observations in captivity [14, 16]. Intra-pack competition for dominant breeding positions in the wild appears to be minimal, as most pack structures involve close-kin family groups, with parents naturally maintaining dominance over their offspring [10, 15]; however, inter-pack competition may be fierce. In Quebec, an intruding male assumed the breeding position at the time of the death of the former breeding male [18]. In Denali National Park, conflict between two packs led to two members of the attacking pack using the other pack's den with two unknown wolves, which may be an example of displacement [19].

Since the reintroduction of wolves to YNP in 1995, events such as a change in breeding position have been observed often due to the exceptional visibility of several wolf packs each year [20]. In some cases, dominant males were killed by a neighboring pack and subsequently one or more of the attackers filled in the breeding vacancy. In other cases, breeding males were displaced from their hierarchical position but remained with the pack as a subordinate. Most of these cases involved related males. Breeding males have also been displaced from their hierarchical position and subsequently disperse. Here we observe one such event in detail. We discuss the importance of this event to long-term pack success. We also discuss female mate choice as the main factor leading to successful displacements of a male breeding position while acknowledging that male-male competition can influence that choice.

The two packs involved in this extended event were the Mollie's pack and the Wapiti Lake pack and both lived in YNP. The Mollie's pack lineage was originally reintroduced into YNP in 1995 and has continued to live in the park's interior to the time of publication [6]. The Wapiti Lake pack formed in the park's interior in 2014, although the dominant female was

born in the area and had been a resident since 2010. The two packs territories were adjacent in some areas and even overlapped slightly (Fig 1). Since the formation of the Wapiti Lake pack in 2014, the proximity of their territory to the Mollie's pack territory contributed to occasional aggressive inter-pack encounters and the two packs certainly knew the scents and howls of one another well through the years.

Wapiti Lake dominant breeding male 755M was displaced over the course of the 2016 summer by three adult males from the Mollie's pack. This event was likely a result of a complex interplay of male-male competition and female mate choice [21, 22]. Several factors likely influenced competition and mate choice in this instance, including male age, the number of males, male size, relatedness between the female adults and male adults, and presence of dependent pups. The aim of this writing was to report on a unique, detailed observation of dominant male displacement in gray wolves and discuss resulting implications. As we will show, this single, stochastic event had a significant impact on the future of not only both packs involved, but also on the pack and genetic structure of the entire YNP wolf study area.

## Methods

### Study system

Observations occurred within YNP, with the majority occurring in Hayden Valley (44.6886°N, 110.4655°W) between 7 July 2016 and 12 August 2016 near the rendezvous site (pup rearing homesite) of the Wapiti Lake pack [23]. The valley is mostly open with some coniferous forest cover and several bodies of water, including the Yellowstone River. Openness allows for easy observation within Hayden Valley. Furthermore, the study area is transected by the park road, which provides access for visitors and researchers and serves as the main platform for observing wolves. Elevations within YNP range from 1500–2400 meters, with Hayden Valley at approximately 2300 meters in elevation [24]. Vegetation in the study area consists of lower elevation montane ecoregion Douglas firs *(Pseudotsuga menziesii)*, Wyoming big sagebrush *(Artemisia tridentae)*, and grasslands dominated by *Festuca* sp. to more closely canopied lodgepole pine *(Pinus contorta)* and subalpine fir *(Abies lasiocarpa)*.

The study area also contains abundant wildlife, including ungulates such as American bison *(Bison bison)* and American elk *(Cervus canadensis)*, and large carnivores, such as grizzly bears *(Ursus arctos)* and American black bears *(Ursus americanus)*. The primary prey of wolves in the study area is elk [25]. An abundance of elk and other prey, including bison, enables a high wolf density in YNP (averaging 56 wolves/ 1000 km$^2$ with fluctuations between 20 and 98 wolves/ 1000 km$^2$ [10]). Over the past decade, wolf densities have remained relatively constant, hovering around 40 wolves/ 1000 km$^2$ [26].

### Ethics statement

All capture and handling of wolves were conducted in strict accordance with approved veterinarian protocols and the National Park Service's Institutional Animal Care and Use Committee (IACUC permit IMR_YELL_Smith_wolves_2012) to ensure safe animal welfare. Telazol was used to anesthetize wolves, which resulted in their immobilization prior to handling.

### Telemetry collar monitoring

The Yellowstone Wolf Project monitors wolves using radio collars deployed by helicopter capture in winter with the objective of maintaining enough collars to track each pack. The Wolf Project consistently monitors wolves through aerial and ground radiolocation and observation. At the time of capture, newly collared wolves are assigned unique identifiers (an ordered-

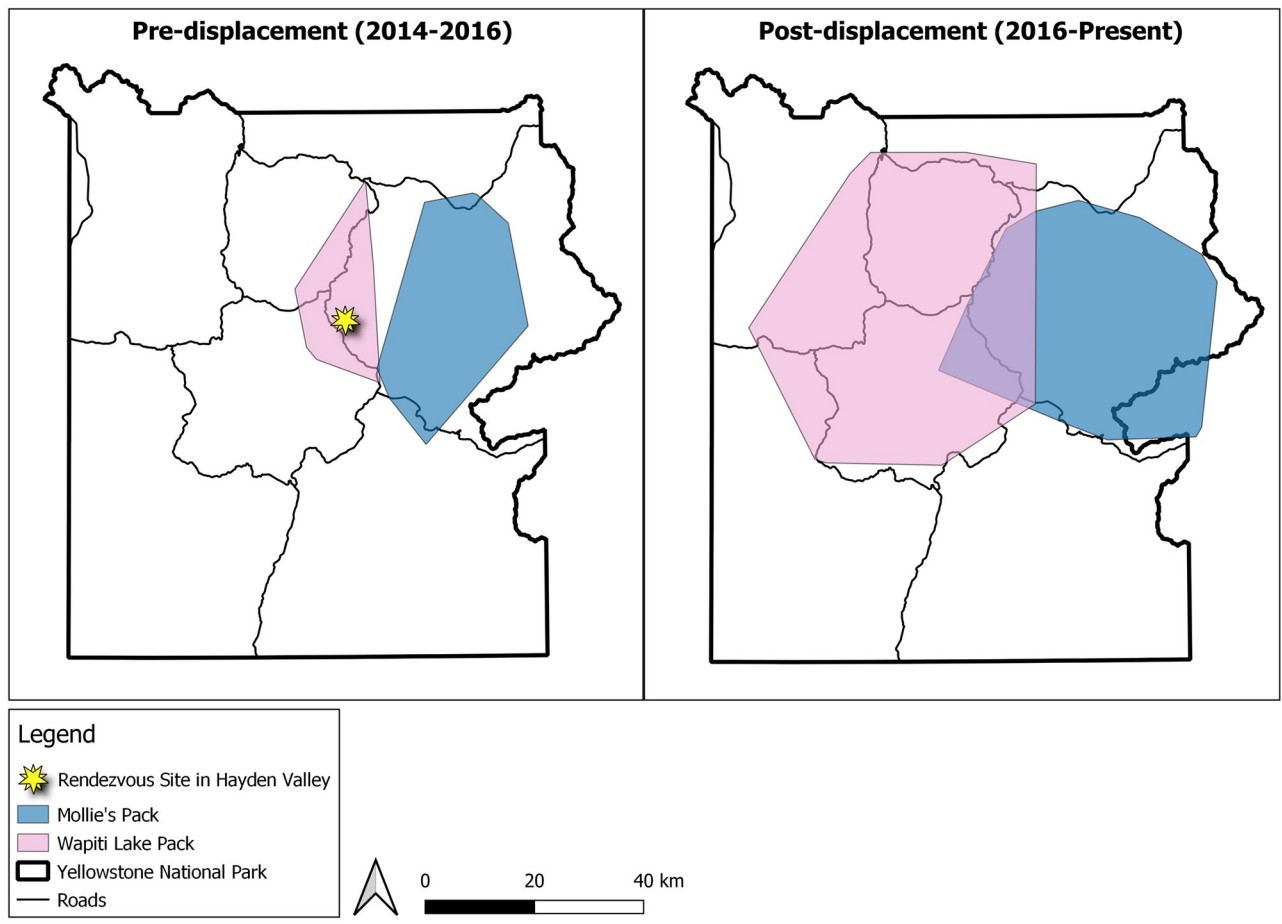

**Fig 1. Wolf pack territory map before and after displacement.** 95% minimum convex polygons of the home ranges of the Wapiti Lake pack and the Mollie's pack prior (2014–2016) to the displacement of the Wapiti Lake breeding male 755M (left) and post (2016 to present) displacement (right) in Yellowstone National Park. Proximity of the two packs' territories likely contributed to the displacement of breeding male 755M, as described in this writing. However, wolf packs in Yellowstone National Park frequently have territories that abut or even overlap significantly. As breeding displacement has been fairly rare, it is unlikely that territory proximity explains the trigger for this event, rather it contributed to the ease of contact between packs.

numeric followed by a sex identifier, e.g., 755M), which are used below to delineate specific collared wolves from uncollared wolves. Both GPS collars and VHF collars are deployed, and wolves are typically located from the air or ground using the VHF signals approximately five times a week. The GPS collars typically take fixes every four hours, and upload every twelve fixes. Telemetry was used to locate wolves and spotting scopes and binoculars were used for observation.

## Observations

Observation notes were recorded on a voice recorder and then transcribed to data forms and a journal. Several individuals were involved in the observation process and collaborated to summarize wolf behavior. All of the adult wolves involved were individually-recognizable by experienced observers based on unique pelage colors and patterns, body size, and collar types. The three gray pups all looked similar and were not individually-recognizable (Fig 2).

## Results

The two packs involved in the encounter were the Wapiti Lake pack and the Mollie's pack. These packs occupied adjacent territories in central YNP, and territorial forays by the Mollie's

| Wolf ID | Pack | Gender | Age | Color |
|---------|------|--------|-----|-------|
| 1014M | Mollie's | Male | 3 | Black (White Box on Collar) |
| 1015M | Mollie's | Male | 2 | Black (Black Collar) |
| Mollie's Male | Mollie's | Male | 2 | Black |
| 755M | Wapiti Lake | Male | 8 | Silver/ Blue |
| Wapiti Lake Dominant Female | Wapiti Lake | Female | 5 | Light gray, turning white |
| Wapiti Lake Yearling | Wapiti Lake | Female | 1 | Gray |
| Wapiti Lake Pups (4 pups) | Wapiti Lake | 3 Female, 1 Male | <1 | 3 Gray, 1 Black |

**Fig 2. Notable wolves involved in the displacement.** Notable wolves involved in the displacement of Wapiti Lake breeding male 755M by males from the Mollie's pack in the summer of 2016. Note the age differences between the males from the Mollie's pack and 755M. Adult wolves were individually-identifiable. Graphics by Kira Cassidy.

pack into the Wapiti Lake pack's territory were common (Fig 1). At the time of the displacement, the Wapiti Lake pack consisted of seven wolves, including dominant breeding male 755M, a white dominant breeding female, a gray female yearling, three gray pups, and one black pup (Fig 2). The Mollie's pack consisted of sixteen adult wolves and four pups. However, the entire Mollie's pack was not involved in this encounter. Rather, three members of this pack were involved, including the three-year old male 1014M, two-year old male 1015M, and a black male two-year old who would later be collared as 1155M. The first known encounter between the Mollie's pack and the Wapiti Lake pack was observed on 7 July 2016. The final observation took place on 12 August 2016, when 755M was observed near the Wapiti Lake pack for the last time. There was considerable variation in the size and age between the male wolves involved in this encounter. At the time of capture, the radio-collared Mollie's males weighed 56 kg and 58 kg at 2 and 3 years old respectively, and eight-year-old 755M weighed 40 kg (Table 1). In male wolves, body mass tends to decrease with age after five years indicating 755M was likely even smaller than his capture weight, which was two years prior to the capture of the Mollie's males [23]. In contrast, the three intruding males were approaching their maximum body size. By leveraging a genetically derived population pedigree [27], we estimated little difference in the males' relatedness to the unsampled dominant female, determined by their relation to her parentage (Fig 3). There was a large difference in relatedness to the gray female yearling. We discuss later how these factors could have contributed to this encounter's outcome.

**Table 1. Weight of notable male wolves.**

| Male Wolf ID | Pack | Weight (kg) | Capture Date | Age at Capture |
|---|---|---|---|---|
| 755M | Wapiti Lake | 40 | 17 January 2014 | 6 |
| 1014M | Mollie's | 56 | 26 January 2016 | 3 |
| 1015M | Mollie's | 58 | 26 January 2016 | 2 |
| 1155M | Mollie's | 50 | 11 December 2018 | 5 |

Weight of notable radio-collared male wolves involved in the displacement of Wapiti Lake breeding male 755M. Note the size differences in the two males from the Mollie's pack and Wapiti Lake male 755M. In addition to being outnumbered, 755M was much smaller than the three invading males. Male wolves tend to lose mass as they age [23], indicating 755M was likely smaller at the date of displacement than his capture date.

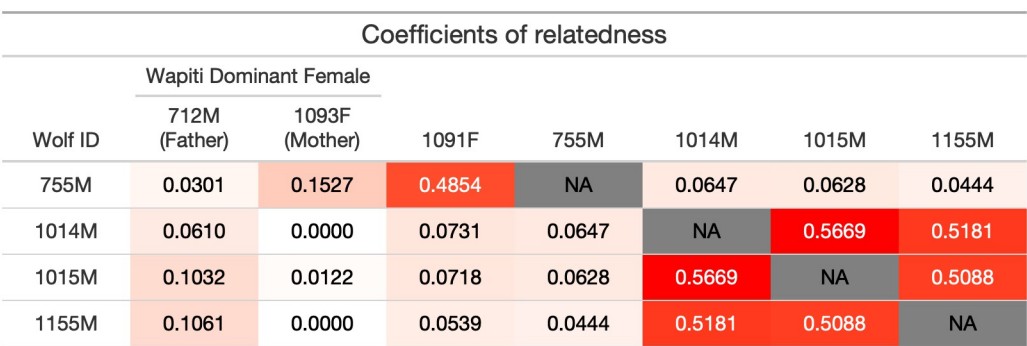

**Fig 3. Coefficients of relatedness for genetically sampled wolves involved in the displacement of Wapiti Lake dominant male 755M in Yellowstone National Park.** Cell colors range from white (relatedness value of 0, indicating a pair is completely unrelated) to red (highest relatedness value of 0.5669, belonging to the sibling pair of 1014M and 1015M). Relatedness of approximately 0.5 is indicative of either full sibling or parent/offspring relationships. The uncollared black male from the Mollie's pack was later captured and assigned the number 1155M. The gray female yearling was also later captured and assigned the number 1091F. The Yellowstone Wolf Project has not genetically sampled the Wapiti Lake dominant breeding female, so her parents' (712M and 1093F) coefficients of relatedness are included here. See vonHoldt et al. 2020 for genotyping methods used to estimate relatedness [27].

## Daily observation summaries

**7 July 2016**: In the morning, seven wolves from the Mollie's pack, including 1014M, who was a black three-year old male, were observed near the Wapiti Lake pack's rendezvous site in Hayden Valley.

**10 July 2016**: At *0543*, three black wolves from the Mollie's pack, including 1014M and 1015M, chased the Wapiti Lake breeding male 755M and the breeding female. From this day on we recognized the three black wolves involved, who were brothers. The other four Mollie's wolves returned to their own territory and did not participate further in this encounter. The Wapiti Lake pair split up and the intruding males chased the female. The breeding pair from the Wapiti Lake pack regrouped with the Wapiti Lake female yearling and the three males from the Mollie's pack moved out of sight. By *1300*, GPS points showed 1014M south of the Wapiti Lake pack's rendezvous site. At *1600*, the Wapiti Lake breeding female and two pups were seen back in the pack's rendezvous site.

**11 July 2016**: Around *1200*, 1014M and another black wolf were seen in the rendezvous site with the Wapiti Lake pack's breeding female and female yearling. This was the first day the Wapiti females were seen interacting with the Mollie's males in a friendly way. The wolves

sniffed each other, and the females jumped on the males and put their heads on their backs. This is typical wolf courting behavior. This behavior was also observed in the evening. The Wapiti Lake breeding male, 755M, was seen at *2230* a few kilometers away from the rendezvous site. No Wapiti Lake pups were seen.

**13 July 2016**: At *1150*, two males from the Mollie's pack and two females from the Wapiti Lake pack were seen in Cascade Meadows (44.7336˚N, 110.5093˚W), a regular hunting area for the Wapiti Lake pack. At *1256*, 755M was seen attempting to cross the road near the rendezvous site. He successfully crossed at *1530*. He was then seen near the rendezvous site with the four pups from the Wapiti Lake pack. GPS points indicated 1014M was in the traditional territory of the Mollie's pack in the evening.

**14 July 2016**: In the morning, 755M was seen in the rendezvous site. At *1430*, the two males from the Mollie's pack and two females from the Wapiti Lake pack were seen in the rendezvous site. They were seen in the same area in the evening. Blood on the breeding female's face indicated the wolves had made a kill. This might be an indication that the females had made their choice regarding which males to stay with as the focus was back to regular wolf life (e.g., hunting) instead of on the intruding males.

**15 July 2016**: At *0810*, 755M and the Wapiti Lake yearling female were seen south of the rendezvous site. At *1100*, the two Mollie's males were seen chasing 755M. The chase was slow paced. The Wapiti Lake females followed the males from the Mollie's pack. The four pursuing wolves disappeared behind a hill and were not seen again. After looking back, 755M went out of sight. At *2015*, the breeding female and two Mollie's males were seen to the south of the rendezvous site.

**16 July 2016**: Around *0900*, the two Wapiti females were seen near the rendezvous site with three Mollie's males including 1014M, 1015M, and the uncollared black two-year old. The Wapiti Lake yearling was seen putting her head over the backs of 1014M and 1015M. The breeding female did this to 1014M. He then put his head over the back of the female. She playfully lunged and nipped at him. Raised leg urinations were seen from all three males, indicating dominance, and both females marked with them, indicating pair-bonding [28–30]. The wolves began to travel south. They encountered an elk herd at *1439*. The chase ended out of sight but was successful. The breeding female was seen alone carrying an elk leg to the north of the rendezvous site early in the evening. She was likely feeding the pups. At *2143*, the breeding female and the three Mollie's males were seen near the carcass.

**20 July 2016**: At *0930*, the two Wapiti Lake females and three males from the Mollie's pack were seen near the rendezvous site. At *1930*, the five wolves were then seen with the four Wapiti Lake pups, sired by 755M, near the rendezvous site. This was the first direct observation of the Mollie's males with the Wapiti Lake pups.

**21 July 2016**: At *0656*, the five adult wolves (two Wapiti Lake females and three Mollie's males) were seen with the four pups. During the morning all the adults were seen greeting the pups. At *0847*, 755M was briefly observed 300 meters east of the other wolves. At *1630*, the wolves were seen again. One of the Mollie's males played with a pup.

**22 July 2016**: At *0736*, the five adults and four pups were seen near the rendezvous site. When 1014M moved toward the pups, they rushed to greet him. The female yearling played with two of the Mollie's males. All three males played with the pups. At *1757*, 755M was seen near the other wolves. Although he got close, neither 755M nor the main pack seemed to be aware of each other. Later in the evening, the three Mollie's males saw and chased 755M at a trot.

**23 July 2016**: After *0700*, all five adults and four pups were seen near the rendezvous site. At *1150*, 755M was seen east of the other wolves. One of the collared Mollie's males slowly

chased him out of sight. In the evening, 755M was seen in the rendezvous site alone. The other five adults were seen to the south chasing elk.

**24 July 2016**: At *0801*, there were brief views of the gray female yearling and some pups in the rendezvous site. The pups came out and played while the yearling went out of sight. At *1015*, 755M appeared and bedded near the playing pups before he approached them. They greeted him. By *1215*, 755M left and crossed the road to the west. At *1405*, the five adults returned to the rendezvous site from the south. The three Mollie's males smelled the area and scent marked. There were brief sightings of the five adults in the evening.

**26 July 2016**: At *0752*, two of the pups were seen. Around *0900*, the five adults and two other pups were seen near the rendezvous site. Shortly after this, 755M appeared in the area and walked by the breeding female. They did not seem aware of each other as neither looked in the direction of the other. He moved out of sight at *0930*. Later, the three Mollie's males followed his scent into the trees. The Mollie's males reappeared in a gap in the trees with two of the pups and the yearling female. They greeted each other at *0944*. The breeding female followed their scent. At *0946*, 755M appeared again moving the opposite direction. He passed all the other wolves without being noticed. At *0955*, he had a tucked tail, flat ears, and was glancing around in all directions. He moved out of sight. The other wolves appeared on the same trail moving quickly in his direction. They moved out of sight in the same area where 755M disappeared. They came back into view and moved back to the rendezvous site. The uncollared Mollie's male smelled the area where 755M was with a raised tail at *1056*. At *1100*, 755M was seen crossing the road away from the rendezvous site. In the evening, the pups and five adults were seen in the rendezvous site.

**27 July 2016**: Around *0842*, the five adults were seen west of the road from the rendezvous site. They went out of sight moving west. There were brief sightings of the pups in the rendezvous site. At *1615*, 755M was seen in the rendezvous site with the breeding female and at least three of the pups. The yearling and the three Mollie's males were not seen.

**28 July 2016**: At *0852*, three pups and the yearling female were seen in the rendezvous site. An aerial wolf tracking flight found 755M about three miles north of the rendezvous site. At *1315*, the breeding female crossed the road from the west and went to the rendezvous site. She fed and played with all four pups. At *1400*, the yearling female and the three Mollie's males unsuccessfully attempted to cross the road from the west. At *2037*, the breeding female, yearling female, pups, and 755M were seen in the rendezvous site. The three Mollie's males were not seen.

**29 July 2016**: By *0700*, the three Mollie's males, breeding female, female yearling, and the four pups were seen in the rendezvous site. A gray pup greeted 1014M. The wolves moved out of sight. Around *1905*, the adults chased a bison calf. The pups were seen in the rendezvous site. At *1945*, 755M emerged from the trees and the pups greeted him. He was visibly nervous, exhibiting a tucked tail and flattened ears, but remained with the pups until dark.

**31 July 2016**: Before *0800*, the nine wolves were seen in the rendezvous area. At *0924*, 1014M and 1015M began running towards something. Former breeding male 755M came out of a gully but quickly retreated as the two Mollie's males ran toward him. The two Mollie's males went out of sight behind him. All the wolves eventually went out of sight in that area. Around *1908*, 755M appeared, but was chased by the three males as the female went to the pups. The uncollared male was less interested, but 1014M and 1015M continued the chase. Abruptly, 755M stopped and stood his ground. Although contact could have been made, none was observed. The males backed off and all the wolves bedded approximately 50 meters away from 755M.

**2 August 2016**: At *0751*, the five adults were seen in the rendezvous site. The pups eventually appeared. Former breeding male 755M appeared and two of the Mollie's males began to

trot toward him. He bedded and howled. The Mollie's males also bedded. At *0803*, 755M got up and moved away, looking back often. The Mollie's males got up and followed him before bedding again. He continued uphill. He bedded at *0813*. He got up and moved out of sight at *0832*. He came back out at *0910* and looked at the other wolves. He bedded and continued to watch them. He eventually got up and moved out of sight after rolling in an area where the other wolves scent marked. At *1900*, 755M was seen coming out of the trees and he then bedded in the rendezvous site. The other adults appeared, and he quickly moved off. Two of the pups ran toward him. One of the Mollie's males chased him off slowly. The pups and the adults turned and disappeared into the trees. He followed them at a distance.

**3 August 2016**: By *0752*, the nine wolves were seen in the rendezvous site. The adults moved to a nearby bison carcass. At *0927*, 755M appeared to the north of the other wolves. He went out of sight. At *1800*, 755M was seen in the rendezvous site with the yearling female and the pups. The other four adults arrived from the east and 755M got up and approached them. One black slowly chased 755M. The chase quickly ended, with all the wolves bedding. The three Mollie's males and 755M bedded about 75 meters apart by *2020*.

**4 August 2016**: At *0814*, the four pups were observed playing. The yearling female and 755M were bedded nearby watching them. The wolves moved out of sight by *1125*. At *1940*, the pups came out of the trees. The yearling and 755M reappeared as well. Suddenly, 755M raised his tail and ran towards the pups in an aggressive manner. The pups ran into the trees, but quickly returned and licked 755M's muzzle seeking food. The pups then moved to the yearling female and licked her muzzle. The two adults bedded down, and the pups played. They moved out of sight by *2022*. At *2145*, the breeding female crossed the road and returned to the rendezvous site. The three males from the Mollie's pack were unable to cross due to traffic.

**5 August 2016**: Early in the morning, the five adults and the four pups were seen in the rendezvous area. At *0910*, 755M was seen near the road. He had fresh puncture marks in his thigh and his fur had blood on it. He was limping, but the wound was not severe. He moved out of sight at *1025*. The other nine wolves were seen in the rendezvous site in the evening.

**12 August 2016**: At *0708*, the four pups were seen in the rendezvous site. A flight located 755M moving toward the rendezvous site. After he checked the area, the pups saw him and ran to him. They greeted him, seeking food. At *1015*, the pups and 755M greeted each other again. They moved out of sight. At *1059*, the other adults approached the rendezvous site. The pups came out to greet them. At *1150*, 755M swam across Alum Creek and crossed the road away from the other wolves. Only the four pups were seen in the rendezvous site in the evening. Former breeding male 755M was not seen with the Wapiti Lake pack again.

Wolf 755M was not seen with the Wapiti Lake pack again. He was observed twice in Hayden Valley after displacement on 26 August 2016 and 4 September 2016 but was not observed interacting with other wolves. By the winter of 2016–2017, 755M joined the Beartooth pack (approximately 60 kilometers NW of Hayden Valley), was observed breeding within the pack, and may have fathered a litter of pups before he went missing in spring 2017. The three males from the Mollie's pack remained with the Wapiti Lake pack females and pups. 1015M became the dominant breeding male and bred with the dominant female and the yearling female in 2017 and the dominant female and another yearling female in 2018. The pack produced twelve pups in two litters in 2017 and seven pups in two litters in 2018, totaling at least nineteen pups, before 1015M dispersed and joined a different pack (along with the uncollared male). Subordinate male 1014M then became the dominant breeder in Wapiti Lake and bred with the dominant female and another adult female in 2019 and 2020. The pack produced nine pups in two litters in 2019, eight pups in two litters in 2020, and at least ten pups from two litters in 2021. This displacement has resulted in at least forty-six pups produced at the time of this publication.

## Discussion

Breeder displacement events in social carnivores are rarely observed, and most observations of displacement either result in mortality or expulsion of the breeding individual within a short period of time. This case is unique for multiple reasons. First, the length of the displacement was exceptionally long and complete to our knowledge. However, because wolves are not always in observable viewsheds and can be active at night, there were likely several interactions that were not observed. Second, the displacement was relatively amicable. There were no physical attacks observed, and chases were not high intensity. The only indication of a possible violent encounter were puncture marks observed in 755M's thigh in early August. After the initial few weeks, 755M even bedded as close as 75 meters to the intruding males. Third, the presence of young has rarely been documented in displacement events.

On average, about 32% of wolves in YNP are breeders each year (unpublished data, Yellowstone Wolf Project), and when there is a breeding vacancy the position is usually filled quickly. This encounter describes a different strategy of wolves forcing a breeding vacancy in a pack and pack females changing their male-allegiance. This event may have been influenced by the close proximity of the Wapiti Lake and Mollie's pack territories, giving the intruding males easy access to the Wapiti Lake pack's rendezvous site. Although this proximity likely contributed to this displacement case, it was probably not the only factor responsible. The presence of a single male versus three males, the difference in male body sizes and ages, and the choices of the two females likely drove this displacement event and the ensuing results. This event occurred during summer, when inter-pack aggression is typically low [17, 26] and as such was likely heavily influenced by the presence of pups and possibly different hormones compared to the winter months [31].

The presence of pups in this displacement event also provides important context as it may have influenced the probability the event occurred and its outcome. At the time of the first encounter the four Wapiti Lake pups were around 11 weeks old and weighed approximately 12 to 16 kg depending on sex [23, 32]. At this age, the pups were recently weaned off milk but still had deciduous teeth. They relied on the adult wolves to regurgitate or bring meat to them, and their mobility was limited to within and around the rendezvous site, rarely traveling further than a few kilometers from the rendezvous center. This breeding displacement, which resulted in the exchange of an old male for three large prime-aged males, may have had important implications for these pups during a critical period in their growth and development as three prime-aged males could provide more food and protection than one elderly male. The four pups from the Wapiti Lake pack were fed by 755M and the breeding female during the initial stages of the displacement but were quickly accepted by the intruding males, although we never recorded the males regurgitating or bringing food to the pups.

The intruding males from the Mollie's pack chased 755M several times in July and August. The chases were of low intensity and aggression (e.g., running at only a trot or lope), possibly because the invading males knew that 755M did not pose a threat, as he was much smaller and older than the Mollie's males (Table 1). This relatively passive response might be expected if the wolves were related, however, they were not significantly close relatives (Fig 3). Old wolves are also known to be important in territorial conflict, likely due to their knowledge in avoiding dangerous situations [10] and breeding male 755M may have survived these encounters due to his age and experience. In total, there were six chases observed. In two cases, 755M remained near the other wolves after the chase (Table 2). In one case he bedded near them. There were also five times 755M entered the rendezvous site and greeted his pups and former mate when the Mollie's males were not present.

**Table 2. Aggressive encounters between 755M and the intruding male from the Mollie's pack.**

| Date | Encounter Details | Mollie's Wolves Present | Wapiti Lake Wolves Present | Interaction Outcome |
|---|---|---|---|---|
| 10 July 2016 | Three Mollie's males chase the Wapiti breeding pair | 1014M, 1015M | 755M, Breeding Female | Wapiti wolves regrouped |
| 15 July 2016 | Two Mollie's males chase 755M | 1014M, 1015M | 755M | 755M fled |
| 22 July 2016 | Three Mollie's males chase 755M | 1014M, 1015M, Black Male | 755M | 755M fled |
| 23 July 2016 | One Mollie's male chases 755M | 1014M or 1015M | 755M | 755M fled |
| 31 July 2016 | Three Mollie's males chase 755M | 1014M, 1015M, Black Male | 755M | 755M bedded near Mollie's males |
| 2 August 2016 | Mollie's males chase 755M | Two black wolves | 755M | 755M bedded near Mollie's males, then followed them |

Aggressive encounters observed in his displacement as breeding male of the Wapiti Lake pack by three males in the Mollie's pack in the summer of 2016 in Yellowstone National Park. Initially, 755M fled when the intruding males chased him, but toward the end of the displacement event, he was comfortable enough to bed near the males, and in the last encounter even followed them. Despite this behavior, he eventually dispersed and did not return to the pack.

The behavior of each of the wolves involved changed throughout the course of the displacement event. Initially, the breeding female of the Wapiti Lake was chased by the intruding males. Within a week of the chase, both Wapiti Lake females displayed courtship behavior with the males and made a successful kill with them. The four Wapiti Lake pups accepted the intruding males from the beginning of the interaction. The intruding males initially chased 755M frequently, but as the events progressed, the chases became less intense (Table 2). Former Wapiti Lake breeding male 755M initially fled from the intruding males, but in the last few encounters of the displacement, he bedded in close proximity to them. However, he ultimately left the Wapiti pack as the females seemingly preferred the Mollie's males and to remain in the pack with three unrelated males was likely not a comfortable or safe option.

Infanticide is highly unusual in wolves [17, 33], unlike the pattern for newly dominant African Lion *(Panthera leo)* males, which kill lion cubs [6, 34]. Due to wolves only breeding once a year, and female wolves not having spontaneous estrus following lost offspring, the Mollie's males were likely not motivated to kill the dependent pups in this event. The pups may have been allowed to live because they added numbers to the pack size and pack size is important to many aspects of long-term pack success [7, 8, 26, 35]. In addition, allowing the Wapiti pups to live might have been a strategy to find future breeding partners (three of the pups were females). In fact, one of the gray pups bred with at least one of the intruding males in 2018, 2019, 2020, and 2021.

There were two cases in which 755M stayed near the other wolves after being chased. He was also observed standing his ground and approaching the other males during these encounters. This may have been an attempt to rejoin the pack as a subordinate member.

In another instance, 755M visited the rendezvous site and chased the Wapiti Lake pups aggressively before he realized they were his own pups. Wolves have been observed momentarily not recognizing their packmates in territorial conflict, and this may have occurred in this instance. This behavior could indicate that 755M was waiting until one of the other males was alone before attempting to regain his position through aggression. Although it is uncertain why 755M remained in the area so long, there is some indication that he was looking to either regain his breeding position, rejoin the pack again in a subordinate role, or continue providing for his offspring.

Typically, displaced males are either violently displaced through mortality or violently displaced through expulsion from the pack. In rare instances, males have been allowed to remain with the pack following displacement. In YNP, these cases frequently involve related wolves. In the Wapiti Lake case, 755M was not related to the intruding males from the Mollie's pack, and this may have been why they did not allow him to remain with the pack.

Female choice is another important aspect of successful displacement since the behavior of the females from Wapiti Lake made it clear they preferred the Mollie's males over 755M by day four, when they displayed playful behavior toward them and started to spend most of their time with the new males instead of 755M. The relationship between male-male competition and female mate choice is poorly understood, and the interaction between the two is a continued subject of debate [21]. Female choice is also understudied in mammalian species [22] and is likely evolutionarily adaptive. In this case, the Wapiti Lake pack went from having one male adult to having three male adults. In addition to the number of males, their size and age may have been a factor. The three radio-collared Mollie's males were larger (at capture) than 755M (Table 1). They were also younger and at the prime age for hunting large prey (two and three years old), whereas 755M was past prime hunting age at eight years old [36]. This change likely provided greater breeding success and protection of resources and pack members. Successful inter-pack conflict is largely driven by the relative number of wolves in a pack, the number of old individuals in a pack, and the number of large males in a pack [10]. Therefore, the Wapiti Lake pack with the three Mollie's males had the advantage over a scenario where 755M remained the only male. However, 755M was older than the three males. This factor is important because intraspecific conflict, where packs with older wolves are more successful, is the leading cause of death for wolves in YNP [12]. Wolf pup production is maximized when a pack reaches eight wolves [9] and the exchange of three males for shifted the Wapiti pack closer to that ideal size. Large males are also important in subduing large prey, such as elk and bison [36], and litter survival is positively correlated with increasing number of prime age (2–6-year-old) males [6]. By choosing three large, prime-age males, the breeding female may have ensured that her pack would have the advantage in territorial conflict, pup-raising, and hunting large prey, thereby furthering survival. During the summer, the season in which this displacement event occurred, the density and composition of prey are typically homogenous in Hayden Valley, mostly consisting of elk and bison. However, in the winter, most prey animals migrate to lower elevations in northern section of the park. Due to this, the density and composition of prey likely did not impact the choice of the breeding female to stay in Hayden Valley, rather the tradition of her home territory, the presence of her young pups, and the added benefits of pack size were more likely drivers of her decision. In most mammal species, intergroup conflict is biased toward males, however, this does not rule out female involvement. Females of several social species participate in intergroup conflicts, and female choice is likely evolutionarily adaptive in such conflicts [37].

Additionally, the females' acceptance of the intruding males was likely influenced by male-male competition wherein the three Mollie's male were aggressive towards 755M and stayed in the Wapiti Lake pack's territory. It is possible the females experienced some pressure to accept the new males if they wanted to remain in their multigenerational territory as well as protect and raise their non-mobile pups. At a different time of year, without stationary offspring, they may have made a different choice. However, all behavior by the females indicated that by day four they were enthusiastically treating the Mollie's males like pack mates and future breeding partners (e.g., playing, play-bowing, jumping on each other with wagging tails, etc.).

Choosing the three Mollie's males likely resulted in greater reproductive success for the Wapiti Lake pack females. Female wolves incur greater cost related to reproduction than males, likely causing females to select mates more carefully [6]. The three males were unrelated

to both the breeding female and the female yearling, giving both females the opportunity to breed and produce pups. Wolves generally avoid breeding with close relatives [13] and 755M was the father of the gray female yearling. Therefore, the only female 755M could breed with was the dominant female, which is the typical breeding situation in an average wolf pack. However, in the new pack structure the yearling female and the three female pups (once they reached sexual maturity) could breed with any of the three new male pack mates and did not have to disperse or find a temporary mate during the breeding season.

The Wapiti Lake pups from 2016 were accepted into the pack in the long-term, and as of 2021 at least one of the gray females remains with the pack as the beta female. She has since been observed breeding and has produced pups. The other three pups lived with the pack until at least 20 months old and then their fate was unknown. In 2017, both adult females in the Wapiti Lake pack produced litters that were sired by at least one of the three males from the Mollie's pack. Between the two females, twelve pups were born and survived to the end of year. The Wapiti Lake pack became the largest pack in YNP, with twenty-one individuals. Several pups produced in 2017 joined or formed packs of their own once they reached dispersal age, thereby passing on the genes of the Wapiti breeding females and the Mollie's males. As of 2021, five packs in the Greater Yellowstone Ecosystem have breeding members that were born into the Wapiti Lake pack after the 2016 displacement event (Fig 4) a rare and exceptionally successful spread of genes. Conversely, of the eight pups produced by 755M during his two-year tenure as the dominant male of the Wapiti Lake pack, only five survived to their first winter, and none were known to have formed new packs. The only offspring of 755M (from Wapiti Lake) to breed was the gray female yearling breeding with a Mollie's male in 2017.

As of 2021, the Wapiti Lake pack has produced at least 46 pups since the 2016 displacement, all of which were likely sired by one of the three Mollie's males. Up to the end of 2021, 44 of the 46 pups survived to the end of the calendar year in which they were born, for a survival rate of 95.7%. In 2017, all twelve pups produced survived, in 2018, six of the seven pups produced survived, in 2019, all nine pups produced survived, and in 2020, all eight pups produced survived. In 2021, the pack produced at least ten pups, and nine survived to the end of the year. Comparatively, the typical mid-winter survival rate of wolf pups in YNP is around 70% [12]. This reproductive success would likely not have been observed had 755M remained the breeding male of Wapiti during the same years. If 755M remained with the pack, only the breeding female would be likely to breed and produce pups. In the very unlikely event 755M had even lived for five more years he may have produced an average litter of 4–5 pups each year [9] for approximately 23 pups. With a smaller pack to feed and protect them, survival would likely have been average at best, for approximately 16 pups recruited (in comparison to the 44 pups recruited by Wapiti). The intruding males could breed and produce pups with the breeding female, yearling female, and the three female pups. Therefore, the intruding males also benefitted from the displacement, moving from subordinate roles in their natal pack to more dominant roles. Due to this favorable situation, they were not observed interacting with other packs in YNP during the displacement. However, older wolves can sometimes have out-sized positive impacts on their pack due to their experience and accumulated knowledge, especially in inter-pack fights [10], so 755M would have been valuable to the pack in some ways as well.

The causes and consequences of reproductive competition and mate choice in wolves are not well understood but are an important aspect to wolf fitness and evolution. Breeding tenures are often short, and given that wolves have short lives, selection for traits that aid in attaining breeding positions is predicted to be strong [6]. The detailed observation of breeder displacement, subsequent offspring rearing, and new breeding pair formations described here advances our understanding of wolf life history and mate choice strategies among other social

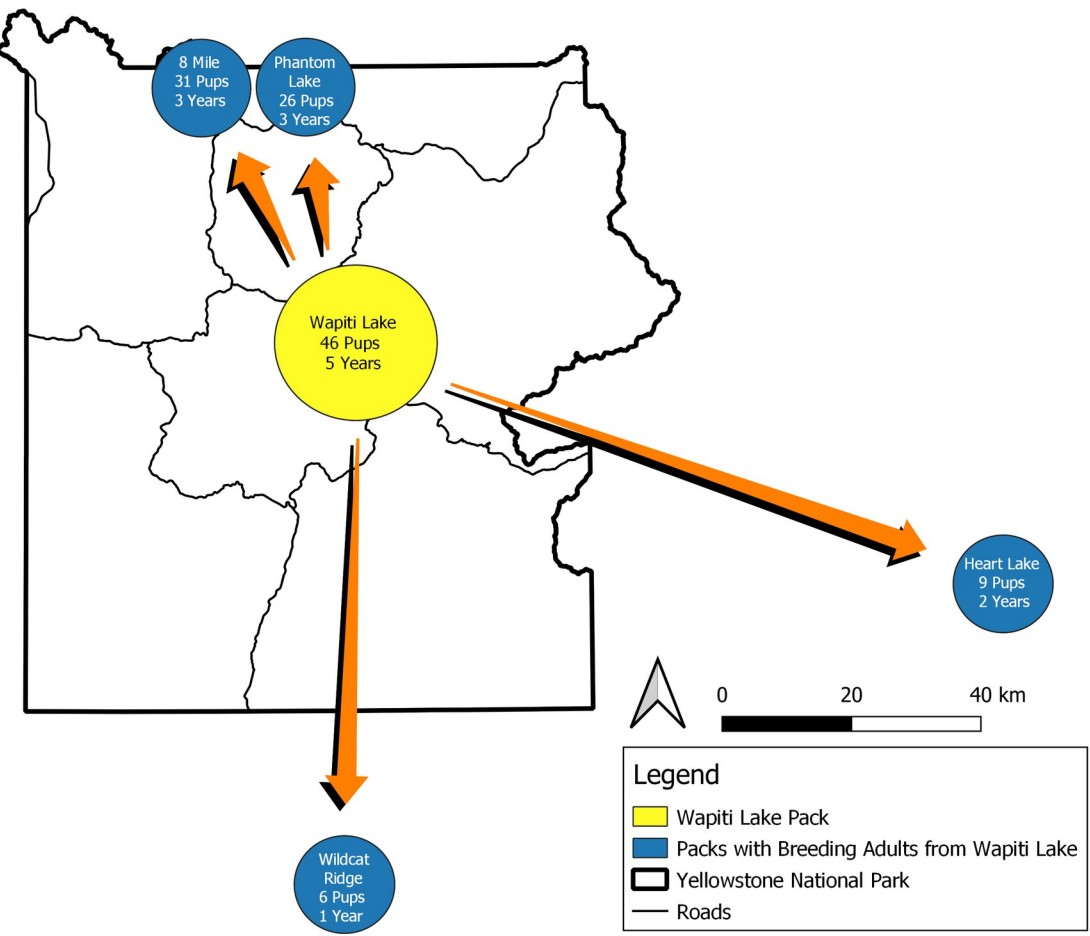

**Fig 4. Resulting wolf pack map.** Gray wolf packs in the Greater Yellowstone Ecosystem with breeding individuals descended from the Wapiti Lake pack after the displacement of dominant male 755M in the summer of 2016. As of 2021, at least four packs in the ecosystem had dominant breeding individuals that were descended from one of the Wapiti Lake females and one of the Mollie's males. Pups produced post-displacement are listed in parenthesis below pack names. These numbers are cumulative pups from five years of reproductive effort (2017–2021).

mammals. As wolves and other wildlife observations continue, more conclusions may be made about mating strategies and competition for breeding positions. Further research into the nuances of breeding displacements, such as the ramifications of these events on pack stability, mate choice, and reproductive success, will help fill an important knowledge gap in our understanding of social dynamics in territorial, social carnivores. Lastly, this displacement highlights the important but sometimes overlooked role of stochastic events in shaping the future of animal populations, even if those events may initially or seemingly affect only a small subset of a population.

## Supporting information

**S1 File. Day by day field notes describing wolf observations in Yellowstone National Park in July 2016.**
(DOCX)

**S2 File. Day by day field notes describing wolf observations in Yellowstone National Park in August 2016.**
(DOCX)

## Acknowledgments

We wish to thank Mark Hebblewhite and Matt Metz for reviewing early versions of the manuscript and for data management. We thank Elizabeth Carroll for data collection. We also wish to thank Joe Bump for recommendations on writing and figures. The knowledge of Wyoming Fish and Game Department staff members Ron Blanchard and Ken Mills were instrumental to this work. We appreciate the cooperation of Laurie Lyman, Doug McLaughlin, and Kathie Lynch who were willing to share their observations of the interactions between the Wapiti Lake pack and the Mollie's pack. Thanks to Rebekah SunderRaj for manuscript review and editing.

## Author Contributions

**Conceptualization:** Jeremy SunderRaj, Jack W. Rabe, Kira A. Cassidy, Douglas W. Smith.

**Funding acquisition:** Daniel R. Stahler, Douglas W. Smith.

**Investigation:** Jeremy SunderRaj, Kira A. Cassidy, Rick McIntyre, Douglas W. Smith.

**Methodology:** Jeremy SunderRaj, Jack W. Rabe, Rick McIntyre.

**Visualization:** Jeremy SunderRaj, Jack W. Rabe, Kira A. Cassidy.

**Writing – original draft:** Jeremy SunderRaj.

**Writing – review & editing:** Jeremy SunderRaj, Jack W. Rabe, Kira A. Cassidy, Rick McIntyre, Daniel R. Stahler, Douglas W. Smith.

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
