## [Decision Letter · Decision Letter 0]

20 Apr 2022

PONE-D-21-25812Breeding displacement in gray wolves (Canis lupus): Three males usurp breeding position and pup rearing from a neighboring pack in Yellowstone National ParkPLOS ONE

Dear Dr. SunderRaj,

Thank you for submitting your manuscript to PLOS ONE. After careful consideration, we feel that it has merit but does not fully meet PLOS ONE’s publication criteria as it currently stands. Therefore, we invite you to submit a revised version of the manuscript that addresses the points raised during the review process.

I would like to sincerely apologize for the delay you have incurred with your submission. It has been exceptionally difficult to secure reviewers to evaluate your study. We have now received two completed reviews; the comments are available below. Both reviewers have raised significant scientific concerns about the study that need to be addressed in a revision. Reviewer#1 raised concerns that this work is not in scope for PLOS ONE. I would like to clarify that this meets PLOS ONE publication criteria of presenting an original research contribution (https://journals.plos.org/plosone/s/criteria-for-publication#loc-1). Please clarify better your manuscript contributes to the existing literature on this topic. Please revise the manuscript to address all the reviewer's comments in a point-by-point response in order to ensure it is meeting the journal's publication criteria. Please note that the revised manuscript will need to undergo further review, we thus cannot at this point anticipate the outcome of the evaluation process.

We look forward to receiving your revised manuscript.

Kind regards,

Miquel Vall-llosera Camps.

Senior Editor

PLOS ONE

Journal Requirements:

4. We note that Figure 2 in your submission contain copyrighted images. All PLOS content is published under the Creative Commons Attribution License (CC BY 4.0), which means that the manuscript, images, and Supporting Information files will be freely available online, and any third party is permitted to access, download, copy, distribute, and use these materials in any way, even commercially, with proper attribution. For more information, see our copyright guidelines: http://journals.plos.org/plosone/s/licenses-and-copyright.

5. We note that Figures 1 and 3 in your submission contain map images which may be copyrighted. All PLOS content is published under the Creative Commons Attribution License (CC BY 4.0), which means that the manuscript, images, and Supporting Information files will be freely available online, and any third party is permitted to access, download, copy, distribute, and use these materials in any way, even commercially, with proper attribution. For these reasons, we cannot publish previously copyrighted maps or satellite images created using proprietary data, such as Google software (Google Maps, Street View, and Earth). For more information, see our copyright guidelines: http://journals.plos.org/plosone/s/licenses-and-copyright.

a. You may seek permission from the original copyright holder of Figures 1 and 3 to publish the content specifically under the CC BY 4.0 license.  

Reviewers' comments:

Reviewer's Responses to Questions

**Comments to the Author**

1. Is the manuscript technically sound, and do the data support the conclusions?

Reviewer #1: No

Reviewer #2: Partly

2. Has the statistical analysis been performed appropriately and rigorously? 

Reviewer #1: No

Reviewer #2: N/A

3. Have the authors made all data underlying the findings in their manuscript fully available?

Reviewer #1: Yes

Reviewer #2: Yes

4. Is the manuscript presented in an intelligible fashion and written in standard English?

Reviewer #1: Yes

Reviewer #2: Yes

5. Review Comments to the Author

Reviewer #1: The causes and consequences of reproductive competition and mate selection in wolves are not well understood but are an important aspect to wolf ecology. The information presented in the manuscript is critical as breeder displacement events and rearing of previous male pups by the new usurpers are rarely observed in the wild. The detailed observation of displacement mating pair, rearing of offspring, and new breeding pair formations described by the authors in the manuscript would advances our understanding of wolf behavior. However, the manuscript has been rejected from PlosOne as it not falls under the aim and scope of the journal.

The manuscript provided a critical event regarding usurping of dominant male by another males. So, I encourage the authors to submit the manuscript in the journals where natural history observations are being published.

Before submitting the manuscript to other journals, please incorporate the following suggestions and issues.

1. The materials and methodology need to be precisely explained that how the observations were made. Please clearly specify how the presence of concerned wolves was evaluated, whether ground tracking VHF receivers or GPS fixes?

2. Under the results, the daily observation summaries need to be short and precisely describe essential events rather than day-to-day events.

3. The authors should revise the language to improve readability. There are minor typographical errors throughout the manuscript; please check.

4. While the study appears to be sound, the language is unclear or sometimes over-explained, making it difficult to follow.

5. The authors should try provide the day-to-day observation in tabular or graph form so that it could be easy to understand.

6. The authors are off to a good start; however, this study requires additional information that I believe is lacking, particularly the behavioural changes of involved individuals with the time. There are interactions between new males and Wapiti Lake pack’s females and new males and alpha males, this information is critical for the study and needs to be clearly and precisely explained.

Reviewer #2: Thanks for giving me the opportunity to review SundarRaj et al. manuscript on Male displacement of wolf packs. The manuscript is written nicely and provides vivid description of the natural history observation of the species. I enjoyed reading the manuscript and listed my comments in a separate document.

6. PLOS authors have the option to publish the peer review history of their article (what does this mean?). If published, this will include your full peer review and any attached files.

Reviewer #1: No

Reviewer #2: No

---

## [Author Response · Author response to Decision Letter 0]

3 Jun 2022

5 May 2022

To whom it may concern,

We appreciate the review of our manuscript PONE-D-21-25812” Breeding displacement in gray wolves (Canis lupus): Three males usurp breeding position and pup rearing from a neighboring pack in Yellowstone National Park.” We made nearly all of the recommended changes and explain all changes, additions, and deletions in detail below. 

If you have any questions, please contact the lead author Jeremy SunderRaj at j.sunderraj@hotmail.com or 303-945-5977. 

Sincerely,

Jeremy SunderRaj

Editor Notes

1. The manuscript was reviewed and meets the style requirements of PLOS ONE. All figures were submitted as .tifs or .tiffs with associated titles and legends in the main text. All tables were included using standard word doc formatting.

2. We have attached the minimal dataset from the study as supporting information, containing detailed field notes collected in July 2016 and August 2016. These are listed in the supporting information section as S1 Fig and S2 Fig, respectively.

3. We added a separate section under the Methods which includes a full ethics statement outlining appropriate permitting/approval from the ethics committee (IACUC).

4. We attached a written statement from one of the coauthors, Kira Cassidy, who illustrated the images of the wolves in Figure 2, allowing for their use in the figure. We have also added her name to the figure caption.

5. We removed the base maps in Figures 1 and 3 to address copyright issues. Reviewer 2 also raised the concern that the base maps were confusing, and removing them also addressed this issue.

Reviewer 1

1. Reviewer 1’s first concern regarded how the presence of concerned wolves was evaluated using VHF receivers or GPS fixes. On lines 154-157, we added a description of the collar types used, both GPS and VHF, and included the approximate number of times wolves are located from the ground or the air via the VHF signals, which was around 5 times a week. We also added that the GPS collars typically take 4 fixes a day and upload points every 12 fixes.

2. The second concern from Reviewer 1 was the length of the daily observation summaries. We addressed this by adding a table of significant events on lines 416-423 and limiting the daily observation summaries to aggressive encounters and notable events. We feel that the remaining day-to day observation descriptions are important to describe the unique nature and length of this particular displacement event.

3. Reviewer 1’s 3rd concern regarded typographical errors in the manuscript. All coauthors have re-read the manuscript and have addressed typographical errors throughout. 

4. In similar fashion to concern 3, Reviewer 1 suggested addressing unclear and over-explained language. All coauthors have re-read the manuscript and have made changes to address unclear and over-explained language.

5. Reviewer 1 suggested adding an additional table including key interactions during the displacement event described in the manuscript. We created and added this table on lines 416-423. This is also related to concern 2.

6. Reviewer 1 suggested adding additional information regarding the behavioral changes of the individual wolves involved in the displacement event, so we added a paragraph describing these changes in lines 425-434. This paragraph describes how the females quickly changed their behavior within the first week of the displacement from fleeing from the intruding males to displaying courtship behavior with them and successfully hunting with them, how the Wapiti Lake pups accepted the intruding males throughout the encounters, and how the interactions between 755M and the intruding Mollie’s males decreased in intensity toward the end of the displacement. Despite this, 755M still left the pack and dispersed east of Yellowstone.

Reviewer 2

1. Reviewer 1’s first concern was the lack of a citation following line 56, so we added a citation addressing the following statement: “In cases where male breeding status changes, females may stay with the male that they prefer, with mate selection likely coming from physical, genetic, behavioral, and situational cues.” We used a book chapter by Stahler et al. 2020, discussing wolf breeding strategies. The citation follows: Stahler DR, Smith DW, Cassidy KA, Stahler EE, Metz MC, McIntyre R, et al. Ecology of Family Dynamics in Yellowstone Wolf Packs. In: Smith DW, Stahler DR, MacNulty DR, editors. Yellowstone Wolves: Science and Discovery in the World’s First National Park. University of Chicago Press; 2020 Dec 21. pp. 42-45. https://doi.org/10.7208/9780226728483-011

2. Reviewer 2 suggested rewriting line 56-58, so we rewrote it to say “Mate preference is thought to benefit individuals most likely to bring the greatest direct and indirect fitness benefits.”

3. Review 2’s third concern regarded combining and rewriting lines 58-61 and adding references, so we combined the sentences and included the importance of the benefits of pack size on stability and genetic diversity. References here include papers that found pack size is beneficial in hunting success (MacNulty et al. 2012), pup production and survival (Stahler et al. 2013), and success in conflict between packs (Cassidy et al. 2015). These studies suggest hunting success, pup production and survival, and success in territorial conflict lead to pack stability and genetic diversity, which is noted in the manuscript now. 

4. We added the Latin name, Canis lupus, for the gray wolf on line 62.

5. Reviewer 2 suggested separating lines 72-74, so we separated them into two sentences for clarification. 

6. Reviewer 2 suggested adding citations for captive wolf studies, so we added them on line 78, including Zimen 1976 and Packard et al. 1985. The citations, respectively, follow: 1.) Zimen E. On the regulation of pack size in wolves. Z Tierpsychol. 1976 Jan 12;40(3):300-41. https://doi.org/10.1111/j.1439-0310.1976.tb00939.x and 2.) Packard JM, Seal US, Mech LD, Plotka ED. Causes of reproductive failure in two family groups of wolves (Canis lupus). Z Tierpsychol. 1985 Jan 12;68(1):24-40. https://doi.org/10.1111/j.1439-0310.1985.tb00112.x

7. In response to Reviewer 2’s concern about confusion regarding the background maps in figures 1 and 3, we removed them and left the background white. This also addresses the concerns raised by the editor regarding copyright issues.

8. Reviewer 2 suggested adding a table of the interactions between the intruding males and 755M, so we added a table on lines 416-423. This table describes the interactions between the intruding males and 755M, and gives the results of each encounter. Toward the end of the displacement period, 755M’s reactions to the males chasing him changed from fleeing to bedding near them, and in the final case, following them. This did not stop him from leaving the pack and dispersing east of the park.

9. Reviewer 2 asked for clarification regarding whether captures wolves are immobilized or anesthetized, so we added that wolves are anesthetized using Telazol, which immobilizes them on line 147.

10. Reviewer 2 suggested adding a description of the collar types used and the frequency of location. We added on lines 154-157 that collars are both GPS and VHF, and included the approximate number of times wolves are located from the ground or the air via the VHF signals, which is around 5 times a week. We also added that the GPS collars typically take 4 fixes a day an upload points every 12 fixes.

11. Reviewer two asked if the intruding males interacted with or displaced any other wolf packs in the area. The three intruding males did not interact with any additional wolf packs that we are aware of during the displacement period, and we added two sentences noting this in lines 550-551. The reasoning for this was likely that the intruding males found themselves in an ideal situation with the Wapiti Lake pack, finding several unrelated females to breed with and an older, smaller males who was easily displaced.

12. Reviewer 2 raised a question regarding whether the proximity of the Mollie’s pack and the Wapiti Lake pack was the only reason this displacement happened. We addressed while important, other factors also likely came into play in lines 385-388. Although the displacement certainly was impacted because of the proximity of the two packs, this likely was not the only reason. Other possibilities include the presence of a smaller sized breeding male, the presence of two unrelated females, and the fact that the intruding males outnumbered the breeding male of the Wapiti Lake pack 3 to 1. All of these factors likely contributed to the displacement. 

13. Reviewer 2’s final question addressed whether prey density in the study was homogenous or heterogenous in composition and breakdown and whether that impacted the choice of the breeding female to remain in her territory with the intruding males. We addressed this in lines 483-492. In the summer, prey for wolves is typically homogenous in composition, consisting mostly of elk and bison. Density in Hayden Valley is consistent through the summer, but most prey migrate to lower elevations in the winter. This likely means that her choice to remain in her territory was likely not influenced by prey density and composition, but more likely the age of her pups and the added benefits of pack size that came with the intruding males.

---

## [Decision Letter · Decision Letter 1]

15 Sep 2022

PONE-D-21-25812R1Breeding displacement in gray wolves (Canis lupus): Three males usurp breeding position and pup rearing from a neighboring pack in Yellowstone National ParkPLOS ONE

Dear Dr. SunderRaj,

Thank you for submitting your manuscript to PLOS ONE. After careful consideration, we feel that it has merit but does not fully meet PLOS ONE’s publication criteria as it currently stands. Therefore, we invite you to submit a revised version of the manuscript that addresses the points raised during the review process.

We look forward to receiving your revised manuscript.

Kind regards,

Bi-Song Yue, Ph.D

Academic Editor

PLOS ONE

Journal Requirements:

Reviewers' comments:

Reviewer's Responses to Questions

**Comments to the Author**

1. If the authors have adequately addressed your comments raised in a previous round of review and you feel that this manuscript is now acceptable for publication, you may indicate that here to bypass the “Comments to the Author” section, enter your conflict of interest statement in the “Confidential to Editor” section, and submit your "Accept" recommendation.

Reviewer #2: (No Response)

Reviewer #3: (No Response)

2. Is the manuscript technically sound, and do the data support the conclusions?

Reviewer #2: Yes

Reviewer #3: Yes

3. Has the statistical analysis been performed appropriately and rigorously? 

Reviewer #2: Yes

Reviewer #3: N/A

4. Have the authors made all data underlying the findings in their manuscript fully available?

Reviewer #2: Yes

Reviewer #3: Yes

5. Is the manuscript presented in an intelligible fashion and written in standard English?

Reviewer #2: Yes

Reviewer #3: Yes

6. Review Comments to the Author

Reviewer #2: I sincerely thank the editor for giving me the opportunity to review the paper. The authors have addressed the suggestions I provided and answered the questions satisfactorily.

I would recommend the paper to be accepted.

Reviewer #3: (No Response)

7. PLOS authors have the option to publish the peer review history of their article (what does this mean?). If published, this will include your full peer review and any attached files.

Reviewer #2: No

Reviewer #3: No

---

## [Author Response · Author response to Decision Letter 1]

28 Oct 2022

To whom it may concern,

We appreciate the review of our manuscript PONE-D-21-25812” Breeding displacement in gray wolves (Canis lupus): Three males usurp breeding position and pup rearing from a neighboring pack in Yellowstone National Park.” We made all of the recommended changes and explain all changes, additions, and deletions in detail below. 

If you have any questions, please contact the lead author Jeremy SunderRaj at j.sunderraj@hotmail.com or 303-945-5977. 

Sincerely,

Jeremy SunderRaj

Editor Notes

1. We hyphenated the word interpack to inter-pack on line 102.

2. We changed the incorrect citation number from 9 to 10 on line 141.

3. We changed the incorrect citation number from 9 to 10 on line 411.

4. We hyphenated the word interpack to inter-pack on line 471.

5. We changed the incorrect citation number from 9 to 10 on line 473.

6. We changed the incorrect citation number from 8 to 9 on line 478.

7. We changed the incorrect citation number from 8 to 9 on line 544.

8. We changed the incorrect citation number from 9 to 10 on line 553.

9. We added acknowledgments for Joe Bump for recommendations on writing and figures and Wyoming Game and Fish staff members Ron Blanchard and Ken Mills for their assistance on lines 576-579.

---

## [Editor Report · Decision Letter 2]

3 Nov 2022

Breeding displacement in gray wolves (Canis lupus): Three males usurp breeding position and pup rearing from a neighboring pack in Yellowstone National Park

PONE-D-21-25812R2

Dear Dr. SunderRaj,

We’re pleased to inform you that your manuscript has been judged scientifically suitable for publication and will be formally accepted for publication once it meets all outstanding technical requirements.

Kind regards,

Bi-Song Yue, Ph.D

Academic Editor

PLOS ONE
---

## [Editor Report · Acceptance letter]

7 Nov 2022

PONE-D-21-25812R2 

Breeding displacement in gray wolves *(Canis lupus)*: Three males usurp breeding position and pup rearing from a neighboring pack in Yellowstone National Park 

Dear Dr. SunderRaj:

I'm pleased to inform you that your manuscript has been deemed suitable for publication in PLOS ONE. Congratulations! Your manuscript is now with our production department. 

Kind regards, 

on behalf of

Dr. Bi-Song Yue 

Academic Editor

PLOS ONE